# Two-Stage Unit Tying for Simplifying Differentiable Logic Gate Networks

**Seungheon Lee** [* 1]   **Jeongmin Sun** [* 1]   **Jaeyong Chung** [1 2]

## Abstract

Differentiable logic gate networks map directly to gate-level circuits, enabling ultra-low-latency inference, yet their logic footprint often exceeds FPGA capacity budgets. Tightly fitting a trained model to a target FPGA requires a post-training mechanism to trade off network complexity and accuracy—analogous to pruning in standard neural networks. To this end, we introduce *unit tying*: a simplification that forces selected gates to constants (0 or 1), enabling constant propagation and downstream logic elimination. However, we observe that naively extending pruning criteria to logic networks is unreliable under such near-discrete modifications. We therefore propose a two-stage algorithm for unit tying: (i) a fast Gauss–Newton screening step under a teacher-referenced logit-distortion objective that constructs a high-recall overshoot set and (ii) a refinement step that corrects approximation and interaction-driven errors using a small number of finite-difference evaluations. On CIFAR-10 and MNIST, our method consistently improves the accuracy–area trade-off over common saliency baselines, yielding substantial post-synthesis LUT reductions of up to 48% on CIFAR-10 and 43% on MNIST, with modest accuracy degradation.

## 1. Introduction

Efficient neural network inference has been studied extensively, driven by the need to deploy learned models across diverse compute platforms under tight latency, energy, and memory constraints. A large body of work has focused on reducing arithmetic cost through quantization and binarization, enabling fast execution with bitwise operations on

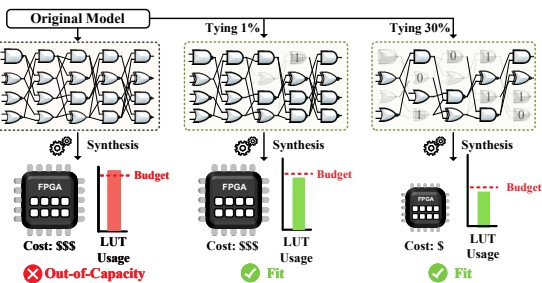

*Figure 1.* Impact of unit tying on FPGA deployment feasibility. While direct synthesis of the trained model results in out-of-capacity failure due to excessive LUT usage, applying unit tying reduces the logic footprint to fit within the target device budget.

commodity processors and specialized accelerators (Rastegari et al., 2016; Umuroglu et al., 2017). More recently, this line of research has progressed beyond low-precision matrix-multiplication pipelines toward *differentiable logic gate networks* (Petersen et al., 2022; 2024), a direction further advanced by recent studies (Yousefi et al., 2025; Fojcik et al., 2025). These networks learn *discrete* gate types from a small operator set (e.g., AND/OR/XOR) by optimizing a *softmax* relaxation over gate choices, and then discretize the learned distribution via *argmax* to obtain a pure gate-level network for highly efficient inference.

A distinctive advantage of logic networks is that, once discretized, their computation aligns closely with the primitives of digital hardware. This makes them attractive for direct FPGA/ASIC mapping, where gate-level structure can be compiled into netlist-like implementations that support very low latency and real-time operation (Umuroglu et al., 2020).

At the same time, hardware mapping imposes a stronger notion of feasibility than software execution: on CPUs/GPUs, oversized models are typically still runnable (albeit slower), whereas on FPGAs even small capacity overruns can prevent successful place-and-route (Umuroglu et al., 2020). This challenge is further amplified in differentiable logic gate networks because the gate count implied by the trained model parameters can differ from the gate count of the finalized circuit after logic synthesis (Petersen et al., 2024), making it difficult to train a network that tightly fits a given FPGA budget *a priori*. Consequently, controlling network size and structure is a first-order requirement for FPGA/ASIC deployment.

*Equal contribution  [1]Department of System Semiconductor Engineering, Yonsei University, Seoul, Republic of Korea [2]Department of Electrical and Electronic Engineering, Yonsei University, Seoul, Republic of Korea. Correspondence to: Jaeyong Chung <jyc@yonsei.ac.kr>.

*Proceedings of the 43rd International Conference on Machine Learning*, Seoul, South Korea. PMLR 306, 2026. Copyright 2026 by the author(s).

In conventional neural networks, pruning is a natural and widely used approach to meet capacity constraints, reducing model size after training while preserving accuracy (Han et al., 2016; Li et al., 2017; He et al., 2017; Molchanov et al., 2017). At a high level, pruning assigns a *saliency* score to parameters (or structured units such as channels) and removes those expected to have the smallest impact on performance. The simplest saliency measure is *weight magnitude*: truncating small weights to zero is often interpreted as introducing a small perturbation with marginal effect on the network function, and it serves as a strong and inexpensive baseline in practice (Han et al., 2016).

More principled pruning criteria define saliency as the true increase in task loss induced by removing or perturbing a parameter, treating this change as a pruning oracle (LeCun et al., 1990). Since evaluating this oracle for every parameter is computationally prohibitive, prior work typically approximates it using a Taylor expansion of the task loss around the current solution. Classic second-order methods such as Optimal Brain Damage (LeCun et al., 1990) and Optimal Brain Surgeon (Hassibi et al., 1993) estimate this increase via Hessian-based approximations, as the first-order term vanishes (or becomes negligible) near a (local) optimum.

In this paper, we present a method to reduce the model size of differentiable logic gate networks, taking into account how these models differ from conventional neural networks. Rather than suppressing or removing parameters—the standard primitive in pruning—we perform unit tying: forcing a unit's output to a constant (0 or 1) to enable constant propagation (Wegman & Zadeck, 1991; Muchnick, 1997) and eliminate downstream logic during synthesis (Chen et al., 2006).

While unit tying departs from pruning in its primitive, the two share the same core question of which units can be modified with minimal impact on model behavior. We therefore adapt the saliency-based pruning perspective. Motivated by function-preservation approaches in model compression (He et al., 2017; Frantar & Alistarh, 2023), we use a teacher-referenced logit-distortion objective for saliency: treating the original trained network as a teacher, we measure the mean squared error between the tied network's logits and the teacher's logits. Unit tying, however, is a discrete and structural intervention rather than a small continuous perturbation, which can violate the assumptions of Taylor-based approximations commonly used in classical pruning.

To address this, we therefore propose a two-stage procedure: (i) a fast screening stage that uses a Gauss—Newton approximation to score candidate units under tie operations, and (ii) a refinement stage that mitigates greedy false positives by efficiently evaluating subsets of candidates via binary-split search. Together, these stages form a distortion-guided tying framework tailored to tie-based compression of differ-entiable logic gate networks, yielding compact circuits that better match FPGA/ASIC resource constraints while maintaining accuracy. Figure 1 illustrates the FPGA feasibility gap and the role of unit tying as a post-training mechanism for meeting device budgets.

**Contributions.**

- To our knowledge, we present the first systematic study of complexity reduction in differentiable logic gate networks via *unit tying*, and identify key differences between pruning and tying in this setting.

- We propose a two-stage tying algorithm that remains effective despite the near-discrete nature of tying actions, combining efficient screening with refinement to handle approximation error and interaction effects.

- We demonstrate that our method consistently outperforms a range of baselines across multiple models on CIFAR-10 and MNIST.

## 2. Related Work

**Logic-centric Efficient Models.** Recent work has explored bit-level logic models to bridge deep learning and hardware primitives. Binary neural networks replace multiplications with bitwise operators (e.g., XNOR–popcount) (Rastegari et al., 2016) and have been deployed via FPGA-oriented frameworks (Umuroglu et al., 2017), while other approaches more directly exploit FPGA soft logic by compiling networks into LUT circuits or learning truth-table operators (Umuroglu et al., 2020; Wang et al., 2020; Benamira et al., 2024). More recently, differentiable logic gate networks learn gate-level circuit topology end-to-end (Petersen et al., 2022; 2024), turning inference into pure logic propagation and achieving substantial improvements over prior state-of-the-art logic-centric models. Because such logic networks are already extremely efficient, they offer limited redundancy, so further compression can naturally incur a larger accuracy penalty than in over-parameterized neural models. We therefore focus on enabling a *post-training* complexity-control mechanism that provides a practical knob to meet diverse hardware budgets.

**Task-Loss-based Pruning.** Most pruning methods were developed for models with continuous parameters, and are commonly guided by minimizing the induced change in task loss. Because selecting an optimal sparse subset is combinatorial and NP-hard (Natarajan, 1995), early pipelines adopt simple proxies such as magnitude-based truncation (Han et al., 2015; 2016). To obtain more principled criteria without combinatorial search, classical second-order pruning estimates the induced task-loss increase from local Taylor/curvature information (LeCun et al., 1990; Hassibi et al.,

1993). Since exact Hessians are costly, more recent work focuses on making curvature-based criteria practical through scalable approximations (Martens & Grosse, 2015; Theis et al., 2018; Singh & Alistarh, 2020). However, such approaches assume local smoothness under small perturbations, which may not hold for discrete functional edits in logic networks. Motivated by this gap, we propose alternative objectives and an efficient procedure tailored to discrete interventions.

**Function Preservation Pruning.** In the context of model compression, some prior works have employed function preservation beyond simply optimizing task loss. One line focuses on *layer-wise reconstruction*, minimizing local feature/output mismatch as a tractable surrogate for the global objective. These approaches cast pruning as a local reconstruction problem and refit the remaining weights accordingly (e.g., He et al. (2017); Frantar & Alistarh (2023)). Another stream leverages Knowledge Distillation (KD) by treating the unpruned model as a teacher and enforcing functional fidelity (Hinton et al., 2015), commonly as an auxiliary signal during post-pruning fine-tuning to recover accuracy (Li et al., 2020). Our work connects these perspectives but takes a different approach: we employ a global function preservation objective directly for pruning saliency, moving beyond local optimization or post-hoc fine-tuning.

## 3. Motivation

In differentiable logic gate networks, computation is carried out by Boolean-like units (i.e., logic gates) arranged in logic layers. Each unit performs one operation from a small, fixed set of 16 logical operations (e.g., AND/OR/XOR), which also includes two constant-output operators. During training, the operator choice of each unit is parameterized by 16 learnable weights; applying a softmax to the weights yields a probability distribution over the operator set. We aim to reduce computation by *tying* selected units to one of these two constants (0 or 1), while minimally altering the model's original function.

Since unit tying is closely related to pruning as a form of structural simplification, it is natural to select which units to tie using the same saliency-based criteria developed for pruning. These criteria typically rely on first- or second-order Taylor approximations, which makes them computationally attractive. In a CIFAR-10 case study comparing ResNet-18 with a medium-width convolutional logic network, we examine whether such criteria can be directly transferred to unit tying.

**(Observation) Unit tying in logic networks can induce substantial changes and is not reliably captured as a small perturbation.** The core assumption underlying

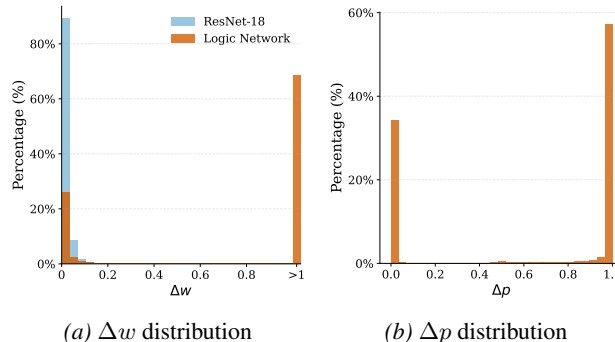

*(a) $\Delta w$ distribution*      *(b) $\Delta p$ distribution*

*Figure 2.* Parameter perturbations required for pruning and tying. (a) ResNet-18 uses absolute magnitude of the zeroed weight, while the logic network uses weight gap between original operator and tying target. (b) The logic network uses the softmax probability gap between the active operator and the tying target.

Taylor-based saliency is that a pruning action induces a *small* perturbation, so that local expansions remain informative. This assumption may be violated by unit tying, which enforces a discrete transition from an input-dependent operator to a constant operator. Figure 2a compares the parameter perturbations ($\Delta w$) induced by weight zeroing in ResNet-18 versus unit tying in a logic network. For the logic network, we define $\Delta w$ as the smaller of the two gaps between the originally selected (argmax) operator's weight and the weights of the two constant operators (0 or 1)—the minimum weight perturbation required to tie a unit. While the perturbations involved in ResNet-18 pruning are relatively small, the $\Delta w$ required for tying in logic networks can be orders of magnitude larger. Figure 2b further shows the distribution of operator softmax probability shifts $\Delta p$ induced when these tying perturbations are applied: roughly 60% of units exhibit $\Delta p \approx 1.0$ —a near-complete reassignment of the selected operator—while the remainder are spread across smaller values. Such discrete-like transitions limit the validity of local Taylor approximations.

This observation motivates a unit tying strategy that combines the tractability of Taylor-based saliency with explicit evaluation of the effect induced by the tying operation.

## 4. Method

We formulate unit tying as a combinatorial optimization problem under a teacher-referenced distortion objective, inspired by function preservation methods (He et al., 2017; Frantar & Alistarh, 2023). To avoid layer collapse, we solve the problem *layer-wise*: for each layer, we select a tying subset $S$ with a fixed budget $|S| = T$ and assign each selected unit to a constant value. We denote this assignment by a mapping $t : S \to \{0, 1\}$, where $t(u) \in \{0, 1\}$ specifies whether unit $u \in S$ is tied to constant-0 or constant-1. In FC-like randomly connected logic layers, we treat every

*Figure 3.* Stage-A uses Gauss–Newton screening to form an overshoot set $S$ of size $T+k$, where $k$ is the overshoot margin. Stage-B isolates a harmful unit via Binary Split refinement (as illustrated), removes the isolated unit from $S$, and repeats this step until $|S|=T$.

unit as a tying candidate. In convolutional logic layers implemented as logic trees, we enforce *structured* tying by restricting candidates to the root unit of each tree, yielding channel-wise tying.

Given a layer-wise configuration $(S, t)$, we measure its functional deviation by

$$\mathcal{L}_{\text{dist}}(S, t) = \mathbb{E}_{x \sim \mathcal{D}} \left[ \tfrac{1}{2} \| z_{S,t}(x) - z_{\emptyset}(x) \|_2^2 \right], \quad (1)$$

$z(\cdot)$ denotes the network output logits (the pre-softmax class scores). While KL divergence is a common choice for distillation-style objectives, we adopt logit-level MSE and empirically justify this choice in our experiments.

Minimizing $\mathcal{L}_{\text{dist}}(S, t)$ directly is intractable due to the combinatorial search space. A simple baseline is to compute a saliency score for each candidate, rank candidates within the layer, and select the top $T$ units. However, as discussed in the previous section, Taylor-based saliency can be inaccurate for unit tying, and evaluating the distortion for every candidate via finite-difference is computationally prohibitive. To address this, we propose a two-stage algorithm (Figure 3). In Stage-A, we use a fast Taylor-based score to screen candidates and construct an *overshoot* set of size $T+k$, where the extra $k$ candidates provide a buffer against approximation errors. In Stage-B, we refine this set by removing $k$ candidates using only a small number of finite-difference (FD) evaluations, yielding a final set of size $T$ with improved reliability.

### 4.1. Stage-A: Second-order Screening via Implicit Jacobian

Although unit tying is a discrete intervention, we analyze its effect by treating it as a perturbation in gate-probability space. For tying unit $u$ to a constant $c \in \{0, 1\}$, let $p_u$ be the gate distribution over operators and $e_c$ the one-hot gate for constant $c$, so that $\delta p_{u,c} = e_c - p_u$. Let $t_c(u) = c$. Leveraging the least-squares structure of Eq. (1), we approximate the resulting distortion via a Gauss–Newton

(GN) quadratic form:

$$\mathcal{L}_{\text{dist}}(\{u\}, t_c) \approx \frac{1}{2} \mathbb{E}_x \left[ (\delta p_{u,c})^\top J_{x,u}^\top J_{x,u} (\delta p_{u,c}) \right]. \quad (2)$$

where $J_{x,u}$ is the Jacobian of the network output logits with respect to $p_u$. In contrast to cross-entropy task loss, where second-order saliency typically involves the parameter Hessian and thus requires additional approximations in practice (LeCun et al., 1990; Hassibi et al., 1993), Eq. (2) depends only on Jacobian products under the distortion loss. This enables a Hessian-free second-order screening rule, which we evaluate efficiently via Jacobian–vector products.

**Jacobian–Vector Products.** While $J_{x,u}$ is far smaller than the full parameter Hessian, it can still be high-dimensional. Following standard automatic-differentiation techniques for Jacobian-vector products (Pearlmutter, 1994; Baydin et al., 2018), we avoid explicitly constructing $J_{x,u}$ and directly evaluate the quadratic form in Eq. (2). Specifically,

$$\| J_{x,u} \delta p_{u,c} \|_2^2 = \sum_{i=1}^d \left( (\delta p_{u,c})^\top \nabla_{p_u} z_i(x) \right)^2, \quad (3)$$

where $\nabla_{p_u} z_i(x)$ is the gradient of the $i$-th logit with respect to the gate distribution $p_u$. In practice, each reverse-mode pass yields gradients for all units in the layer, and per-unit scores are obtained via inexpensive inner products with $\delta p_{u,c}$. Finally, we estimate the GN score $\widehat{s}(u, c)$ by averaging $\frac{1}{2} \| J_{x,u} \delta p_{u,c} \|_2^2$ over $N$ calibration samples.

**Selection.** For each unit $u$, we evaluate both tie directions (0 and 1) and select the less damaging option:

$$\begin{aligned} s(u) &= \min_{c \in \{0,1\}} \widehat{s}(u, c), \\ t(u) &= \arg \min_{c \in \{0,1\}} \widehat{s}(u, c). \end{aligned} \quad (4)$$

We select the units with the lowest $s(u)$ to form an overshoot set $S$ of size $T+k$.

**Algorithm 1** Binary Split Identification

**Require:** Candidate set $S$, tie assignment $t(\cdot)$
1: $S_{\text{curr}} \leftarrow S$
2: **while** $|S_{\text{curr}}| > 1$ **do**
3:    Partition $S_{\text{curr}}$ into two disjoint halves $S_1, S_2$
4:    $\text{err}_1 \leftarrow \mathcal{L}_{\text{dist}}^{\text{FD}}(S \setminus S_2, t)$    *// Tie $S_1$, untie $S_2$*
5:    $\text{err}_2 \leftarrow \mathcal{L}_{\text{dist}}^{\text{FD}}(S \setminus S_1, t)$    *// Tie $S_2$, untie $S_1$*
6:    **if** $\text{err}_1 > \text{err}_2$ **then**
7:       $S_{\text{curr}} \leftarrow S_1$
8:    **else**
9:       $S_{\text{curr}} \leftarrow S_2$
10:    **end if**
11: **end while**
12: **return** the remaining unit in $S_{\text{curr}}$

## 4.2. Stage-B: Refinement via Binary Split

To efficiently identify and remove harmful units from the Stage-A overshoot set, we employ a binary-split search inspired by group testing (Dorfman, 1943). This approach allows us to isolate the most harmful unit using only a logarithmic number of high-fidelity $\mathcal{L}_{\text{dist}}^{\text{FD}}$ measurements, where $\mathcal{L}_{\text{dist}}^{\text{FD}}$ denotes the finite-difference evaluation of $\mathcal{L}_{\text{dist}}$ in Eq. (1) on a calibration set.

Given a candidate set $S$, we partition it into two halves $S_1$ and $S_2$ and perform two FD evaluations under two tie/untie configurations: (i) keep $S_1$ tied while untying $S_2$, and (ii) keep $S_2$ tied while untying $S_1$. We recurse on the half that yields the larger $\mathcal{L}_{\text{dist}}^{\text{FD}}$ under tying. Because $\mathcal{L}_{\text{dist}}^{\text{FD}}$ is evaluated while tying many units simultaneously, Stage-B can also reflect non-additive interaction effects among the units. This single-unit identification procedure is summarized in Alg. 1 and is repeated $k$ times until $|S| = T$, removing one unit per iteration.

**Cached FD evaluation.** To keep refinement scalable, we reuse cached prefix activations up to the current target layer, avoiding full forward recomputation from scratch for each FD evaluation.

**Vectorized evaluation via batch tiling.** To make FD evaluation practical, we evaluate multiple tying configurations in parallel. Given a minibatch of size $B$ and $G$ tying configurations evaluated in parallel, we tile the cached activations to form an augmented batch of size $B \cdot G$, apply configuration-specific tying to each block, and obtain all distortion values from a single forward pass of the suffix network. We repeat this procedure over a calibration set of $N$ samples and average the resulting estimates. For binary-split refinement, we evaluate the two split candidates in parallel (i.e., $G{=}2$) within each forward pass.

**Algorithm 2** Two-Stage Unit Tying

**Input:** Model $f_\theta$, target $T$, overshoot $k$
**Stage-A:** Compute scores $\widehat{s}(u, c)$ for all units $u$
Determine $s(u)$ and $t(u)$ via Eq. (4)
$S \leftarrow \text{Top}_{T+k}(\{u\}; -s)$    *// Select overshoot set*
**Stage-B:** Refine overshoot set
**while** $|S| > T$ **do**
   $u^* \leftarrow \text{BINARYSPLIT}(S, t)$ (Alg. 1)
   $S \leftarrow S \setminus \{u^*\}$
**end while**
Apply ties to units in $S$ using assignments $\{t(u)\}_{u \in S}$

## 4.3. Algorithm and Complexity

The full procedure is summarized in Algorithm 2. We measure computational cost by the number of evaluation passes over the calibration set (each pass averages over $N$ calibration samples).

Let $C$ denote the number of candidate units in the layer. A naive exhaustive strategy evaluates every unit, requiring $\mathcal{O}(C)$ calibration-set forward passes, which quickly becomes prohibitive for wide layers.

- **Stage-A (Implicit Jacobian):** Requires computing gradients of each output logit component with respect to $p_u$. This costs $\mathcal{O}(d)$ calibration-set backpropagations (one per logit per calibration pass), since each backpropagation yields gradients for all $C$ units in the layer simultaneously. The $C$ dependence appears only in inexpensive row-wise inner products. Since typically $d \ll C$, this is substantially more efficient than per-unit finite-difference scoring.

- **Stage-B (Binary Split):** We perform $k$ iterations of binary-split identification. This costs $\mathcal{O}\big(k \log(T{+}k)\big)$ calibration-set forward passes.

## 5. Experiments

### 5.1. Models, Datasets, and Evaluation

We adopt the convolutional differentiable logic gate network architecture proposed by Petersen et al. (2024), which specifies four model variants: `CIFAR-10(M/S)` for CIFAR-10 and `MNIST(M/S)` for MNIST. We train these models and use them for evaluation. All experiments run on a single NVIDIA RTX 4090 GPU. To assess robustness to calibration sampling, we repeat the tying procedure three times with different random seeds and report mean $\pm$ std. Our method uses 16 calibration samples for Stage-A scoring and 80 calibration samples for Stage-B refinement, with an overshoot parameter $k{=}40$. Implementation details for our

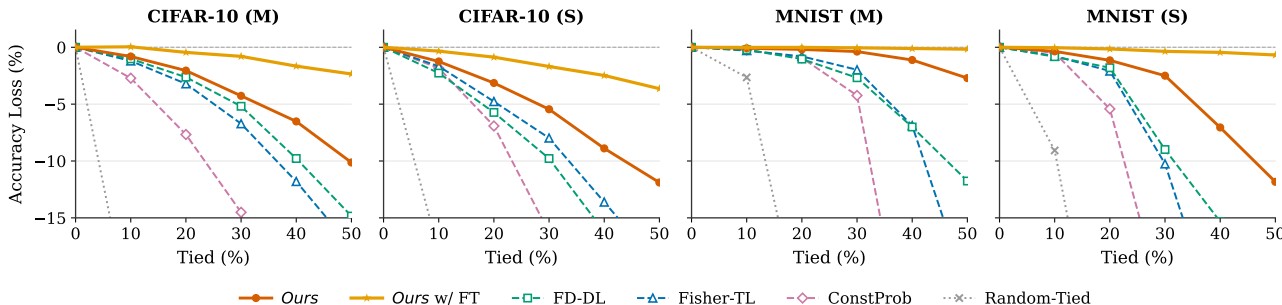

*Figure 4.* Accuracy loss (%) versus tied ratio on `CIFAR-10(M/S)` and `MNIST(M/S)`. Lower is better.

*Table 1.* Top-1 accuracy on `CIFAR-10(M)` and `MNIST(M)` across tied ratios. Bold: best per ratio among without FT and with FT.

| Model | Method | 10% Tied | 20% Tied | 30% Tied | 40% Tied | 50% Tied |
|---|---|---|---|---|---|---|
| `CIFAR-10(M)` (*original*: 71.57) | ConstProb | $68.85 \pm 0.00$ | $63.90 \pm 0.00$ | $57.06 \pm 0.00$ | $32.97 \pm 0.00$ | $24.73 \pm 0.00$ |
| | Fisher-TL | $70.37 \pm 0.24$ | $68.36 \pm 0.49$ | $64.85 \pm 1.02$ | $59.77 \pm 2.57$ | $53.97 \pm 2.92$ |
| | FD-DL | $\mathbf{71.12 \pm 0.06}$ | $69.40 \pm 0.14$ | $67.19 \pm 0.30$ | $64.63 \pm 0.38$ | $56.05 \pm 0.84$ |
| | *Ours (Stage-A only)* | $70.87 \pm 0.24$ | $69.13 \pm 0.23$ | $66.49 \pm 0.26$ | $64.00 \pm 0.33$ | $59.22 \pm 0.97$ |
| | *Ours* | $70.76 \pm 0.18$ | $\mathbf{69.52 \pm 0.25}$ | $\mathbf{67.30 \pm 0.37}$ | $\mathbf{65.05 \pm 0.62}$ | $\mathbf{61.44 \pm 0.53}$ |
| | ConstProb w/ FT | $71.17 \pm 0.10$ | $70.37 \pm 0.08$ | $69.81 \pm 0.22$ | $68.03 \pm 0.12$ | $66.78 \pm 0.11$ |
| | Fisher-TL w/ FT | $71.51 \pm 0.04$ | $70.99 \pm 0.11$ | $70.10 \pm 0.09$ | $69.69 \pm 0.21$ | $68.92 \pm 0.20$ |
| | *Ours* w/ FT | $\mathbf{71.61 \pm 0.03}$ | $\mathbf{71.12 \pm 0.24}$ | $\mathbf{70.77 \pm 0.07}$ | $\mathbf{69.91 \pm 0.07}$ | $\mathbf{69.22 \pm 0.08}$ |
| `MNIST(M)` (*original*: 98.91) | ConstProb | $98.67 \pm 0.00$ | $97.98 \pm 0.00$ | $94.67 \pm 0.00$ | $69.25 \pm 0.00$ | $31.18 \pm 0.00$ |
| | Fisher-TL | $98.68 \pm 0.09$ | $98.08 \pm 0.06$ | $96.64 \pm 0.77$ | $93.94 \pm 1.90$ | $84.69 \pm 5.90$ |
| | FD-DL | $98.75 \pm 0.03$ | $98.48 \pm 0.07$ | $97.76 \pm 0.06$ | $96.38 \pm 0.06$ | $92.18 \pm 1.05$ |
| | *Ours (Stage-A only)* | $98.75 \pm 0.05$ | $98.24 \pm 0.25$ | $96.92 \pm 0.78$ | $95.69 \pm 0.06$ | $91.71 \pm 0.59$ |
| | *Ours* | $\mathbf{98.82 \pm 0.02}$ | $\mathbf{98.72 \pm 0.01}$ | $\mathbf{98.53 \pm 0.08}$ | $\mathbf{97.79 \pm 0.28}$ | $\mathbf{96.20 \pm 0.49}$ |
| | ConstProb w/ FT | $98.85 \pm 0.03$ | $98.85 \pm 0.01$ | $98.77 \pm 0.04$ | $98.54 \pm 0.04$ | $95.12 \pm 0.13$ |
| | Fisher-TL w/ FT | $\mathbf{98.92 \pm 0.02}$ | $98.87 \pm 0.05$ | $98.84 \pm 0.02$ | $98.74 \pm 0.02$ | $98.54 \pm 0.07$ |
| | *Ours* w/ FT | $98.90 \pm 0.02$ | $\mathbf{98.90 \pm 0.00}$ | $\mathbf{98.86 \pm 0.02}$ | $\mathbf{98.80 \pm 0.03}$ | $\mathbf{98.74 \pm 0.00}$ |

baseline re-implementation and post-tying fine-tuning are provided in Appendix A.

### 5.2. Baselines

To validate our method, we compare against several baselines. All baselines apply layer-wise greedy tying: for each layer, they score each unit with a saliency metric and tie a fixed fraction of the least harmful candidates. This procedure is repeated sequentially from early to late layers. The baselines differ only in the saliency metric, as follows:

- **ConstProb.** Ranks each unit by how strongly it already prefers a constant operator, using the larger of the probabilities assigned to the constant-0 and constant-1 operators. This criterion can be viewed as a tying analogue of magnitude-based pruning.

- **Fisher-TL.** Uses an empirical-Fisher second-order proxy (Theis et al., 2018) to estimate the task-loss increase induced by tying, with cross-entropy as the

task loss. This criterion can be viewed as a tying extension of second-order pruning such as Optimal Brain Damage (LeCun et al., 1990). For a fair comparison, Fisher-TL uses the same calibration samples with *ours* ($N=16$).

- **FD-DL.** Evaluates the saliency of each candidate by measuring the finite-difference in the distortion loss. This requires a separate forward evaluation per candidate, making it computationally expensive. To reduce estimator variance, we evaluate it with a large calibration set ($N=128$).

### 5.3. Classification Performance

We apply tying to all layers except the first convolutional layer and the final classifier layer. Table 1 compares our method to all baselines on `CIFAR-10(M)` and `MNIST(M)`. Conventional saliency criteria (ConstProb and Fisher-TL) degrade rapidly as tying increases. In particular, the weaker performance of Fisher-TL suggests that task-loss-based

*Table 2.* Algorithmic metrics (Accuracy, Ops) and hardware feasibility analysis of the `CIFAR-10(M)` model across tied ratios, evaluated on three representative FPGA devices. **Ops** denotes the binary operation count. **Time/Image** denotes the inverse of throughput. **Reduction** denotes LUT reduction relative to the original model. **OOC** indicates Out-of-Capacity (resource budget exceeded). Prices are based on single-unit DigiKey list prices (accessed Jan. 2026).

| Model | Tied Ratio | Accuracy (%) | Ops | Time/ Image | LUT (1000s) | Reduction (%) | FPGA LUT Utilization | | |
|---|---|---|---|---|---|---|---|---|---|
| | | | | | | | V7-2000T ($22,000) | V7-585T ($6,000) | K7-410T ($2,200) |
| `CIFAR-10(M)` | 0% tied | 71.57 | 4.73M | 9 ns | 475.0 | – | 39% | OOC | OOC |
| | 10% tied | 71.68 | 4.44M | 9 ns | 442.8 | 6.77 | 36% | OOC | OOC |
| | 20% tied | 70.47 | 4.05M | 9 ns | 392.4 | 17.39 | 32% | OOC | OOC |
| | 30% tied | 69.63 | 3.66M | 9 ns | 344.6 | 27.44 | 28% | 95% | OOC |
| | 40% tied | 68.64 | 3.22M | 9 ns | 295.3 | 37.82 | 24% | 81% | OOC |
| | 50% tied | 67.06 | 2.80M | 9 ns | 245.4 | 48.33 | 20% | 67% | 97% |

saliency criteria can be less reliable for unit tying than the teacher-referenced distortion objective. At 50% tied, Const-Prob collapses to 24.73% on `CIFAR-10(M)` and 31.18% on `MNIST(M)`, while Fisher-TL improves to 53.97% / 84.69% but still trails distortion-based criteria. FD-DL often outperforms *Ours (Stage-A only)*, indicating that the Taylor-based approximation error can be substantial under unit tying. Despite this, our full method achieves better accuracy than FD-DL in most cases by correcting screening errors. At 50% tied, *Ours* reaches 61.44% / 96.20% on `CIFAR-10(M)` / `MNIST(M)`. With fine-tuning, most of the remaining loss is recovered, bringing aggressively tied models close to the original model. Figure 4 summarizes the accuracy–tied-ratio trade-off; we additionally report a sanity-check reference, Random-Tied, which randomly selects units and tie directions. Results on the smaller variants `CIFAR-10(S)` and `MNIST(S)` are provided in Appendix B.

### 5.4. Hardware Deployment Evaluation

We generate a Verilog hardware description of the trained logic networks and synthesize it with Xilinx Vivado. We follow the implementation strategy of Petersen et al. (2024) and realize the convolutional layers as a spatially unrolled datapath, replicating the per-position logic to maximize throughput. We apply 9 ns (111 MHz) and 5 ns (200 MHz) clock period constraints for `CIFAR-10(M)` and `MNIST(M)`, respectively, enforcing consistent timing constraints and pipeline schedules across all tied ratios. To assess the maximum potential efficiency, we apply unit tying to all layers and report results on fine-tuned models for all hardware experiments. Implementation details and synthesis directives are provided in Appendix A.

To quantify how unit tying trades accuracy for hardware deployability, we target three FPGA platforms of different capacities and measure hardware cost using lookup table (LUT) count, the primary programmable logic resource

in FPGAs. Table 2 reports algorithmic metrics (accuracy and Ops) together with post-synthesis LUT usage and deployment feasibility, where Ops denotes the pre-synthesis binary operation count accounting for constant propagation. Crucially, the original model fits only on the high-capacity FPGA (V7-2000T) and cannot be implemented on the cost-effective medium (V7-585T) and small (K7-410T) FPGAs. However, the significant resource reduction achieved via unit tying successfully enables the design to fit within the strict capacity constraints of the medium FPGA at 30% tying and the small FPGA at 50% tying. This demonstrates that unit tying serves as an effective control knob, trading moderate accuracy degradation for deployability on lower-cost hardware platforms, all while preserving the original ultra-low inference latency.

Table 3 presents the evaluation results for the `MNIST(M)` model synthesized on V7-2000T. Unit tying enables aggressive resource reduction with minimal accuracy drop: even at a 50% tied ratio, it achieves a substantial LUT reduction of 42.88% while incurring a negligible loss of only 0.35%. This further validates unit tying as a generic optimization technique that significantly improves area efficiency. Additional results for the smaller variants, `CIFAR-10(S)` and `MNIST(S)`, are provided in Appendix B.

*Table 3.* Accuracy and LUT usage of `MNIST(M)` model across tied ratios. **Time/Image** denotes the inverse of throughput. **Red.** denotes LUT reduction relative to the original model.

| Model | Tied Ratio | Acc. (%) | Time/ Image | LUT (1000s) | Red. (%) |
|---|---|---|---|---|---|
| `MNIST(M)` | 0% tied | 98.91 | 5 ns | 82.0 | – |
| | 10% tied | 98.91 | 5 ns | 73.1 | 10.81 |
| | 20% tied | 98.90 | 5 ns | 68.0 | 17.00 |
| | 30% tied | 98.92 | 5 ns | 61.8 | 24.57 |
| | 40% tied | 98.69 | 5 ns | 54.1 | 34.00 |
| | 50% tied | 98.56 | 5 ns | 46.8 | 42.88 |

## 5.5. Ablation Study

In this subsection, we validate our design choices through extensive ablation studies on the `CIFAR-10(M)` model.

**Sample efficiency of Stage-A screening.** Table 4 shows the efficiency of Stage-A under varying calibration sample size $N$ at 30% tying. FD-DL is variance-limited at small $N$, but improves as $N$ increases, saturating around $N=64$. However, its wall-clock time grows roughly linearly with $N$ even with vectorized evaluation ($G=32$). In contrast, while *Ours (Stage-A only)* trails the best FD-DL result, adding a short refinement step ($\approx 36\,\text{s}$) already surpasses FD-DL, yielding a better accuracy–time trade-off under limited calibration budgets.

*Table 4.* Sample efficiency of Stage-A at 30% tied: FD-DL (finite-difference baseline, varying $N$) versus our Gauss–Newton screening, reporting accuracy and wall-clock time.

| Method | $N$ | Acc. (%) | Time (s) |
|---|---|---|---|
| FD-DL | 16 | $66.38 \pm 0.35$ | 269 |
| | 32 | $67.23 \pm 0.11$ | 535 |
| | 64 | $\mathbf{67.30 \pm 0.45}$ | 1066 |
| | 128 | $67.19 \pm 0.30$ | 2129 |
| *Ours (Stage-A only)* | 16 | $66.49 \pm 0.26$ | 36.6 |
| *Ours* | 16 | $\mathbf{67.30 \pm 0.37}$ | 72.4 |

**Distortion objective (KL vs. MSE).** We compare Stage-A-only screening performance under different distortion objectives. Table 5 compares KL-based distillation across temperatures $\tau$ to logit-level MSE. KL with $\tau=1$ degrades accuracy, while increasing $\tau$ steadily improves performance and approaches the logit-level MSE result, consistent with the high-temperature limit where KL reduces to squared error on logits (Hinton et al., 2015). Prior work further reports that direct logit-level MSE can be competitive or superior to KL and avoids temperature sensitivity, while small-$\tau$ KL may be preferable under label noise (Kim et al., 2021). Accordingly, we adopt logit-level MSE as our default distortion metric.

*Table 5.* Teacher-referenced distortion objective ablation at 50% tied: KL distillation with temperature $\tau$ versus logit-level MSE.

| Metric | $\tau$ | Acc. (%) |
|---|---|---|
| KL | 1 | $55.88 \pm 0.85$ |
| | 4 | $58.47 \pm 1.49$ |
| | 10 | $58.94 \pm 1.16$ |
| | 20 | $59.14 \pm 1.04$ |
| MSE | – | $\mathbf{59.22 \pm 0.97}$ |

**Refinement Strategies.** We compare our Binary Split (BS) refinement against two refinement baselines: Tail Rescore (TR) and Full-set Rescore (FR). Both methods start from the Stage-A overshoot set of size $T+k$. TR re-evaluates only the bottom $2k$ candidates and removes the worst $k$ units, whereas FR re-evaluates all $T+k$ candidates and removes the worst $k$. Table 6 shows that BS achieves the best accuracy at 30% tied while remaining as fast as TR, and is much faster than FR. This suggests that subset-level FD comparisons in BS efficiently correct approximation-induced errors and can mitigate interaction-driven effects that one-shot per-unit rescoring may miss.

*Table 6.* Refinement strategy ablation at 30% tied ($k=40$). We compare Binary Split to Tail Rescore and Full-set Rescore, which rescore units using FD on a tail subset or the full overshoot set.

| Refinement | Acc. (%) | Time (s) |
|---|---|---|
| Tail Rescore | $66.33 \pm 0.66$ | 35.5 |
| Full-set Rescore | $66.88 \pm 0.20$ | 398 |
| Binary Split | $\mathbf{67.30 \pm 0.37}$ | 35.8 |

**Sensitivity to Calibration Size and Overshoot.** We study the sensitivity of our two-stage procedure to key hyperparameters: the Stage-A calibration size $N_A$, the Stage-B calibration size $N_B$, and the Stage-A overshoot parameter $k$. Figure 5 shows the results of these sensitivity sweeps. Stage-A screening improves as $N_A$ increases up to 16, but changes only marginally thereafter. Stage-B refinement largely saturates at $N_B = 10$, indicating that reliable refinement can be achieved with a modest calibration size. The sweep over $k$ also shows a broad plateau for $k \geq 20$, supporting our default choice $k=40$ as a balance between accuracy and refinement cost. Overall, these results indicate that our two-stage procedure is robust across calibration sizes and overshoot values.

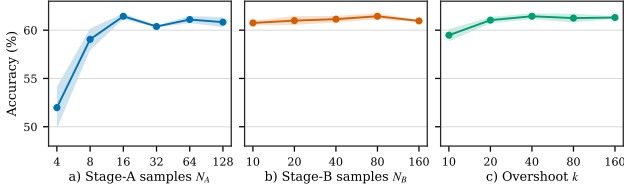

*Figure 5.* Sensitivity analysis at 50% tied. Sweeping (a) Stage-A calibration size $N_A$, (b) Stage-B calibration size $N_B$, and (c) overshoot $k$. Lines show the mean accuracy; shaded bands indicate the standard deviation.

## 6. Conclusion

We study *unit tying* as a synthesis-aware post-training simplification for differentiable logic gate networks. We develop a two-stage procedure using a teacher-referenced distortion

objective, combining fast Gauss–Newton screening with Binary Split refinement. Across CIFAR-10 and MNIST, our method achieves higher accuracy at the same tied ratios and makes previously out-of-capacity designs FPGA-feasible under fixed resource budgets. Overall, our framework offers a practical baseline for bridging logic network training and hardware-constrained deployment. More broadly, tying provides a structured setting to re-examine pruning ideas under near-discrete functional edits, helping separate transferable second-order principles from tying-specific effects.

## Acknowledgement

This work was partly supported by Institute of Information communications Technology Planning Evaluation (IITP) grant funded by the Korea government(MSIT) (No.RS-2026-25525596, Development of Specialized Library Technology Based on On-device AI Semiconductors for Autonomous Agents) and National Research Foundation of Korea(NRF) grant funded by the Korea government(MSIT) (No.RS-2026-25479511, Design of a Next-Generation LLM Accelerator Based on Memory-Centric Data Representations).

## Impact Statement

This work improves the deployability of differentiable logic gate networks under strict FPGA resource budgets by introducing a post-training, synthesis-aware unit tying procedure. By reducing the logic footprint, the proposed method may enable implementation on smaller, lower-cost, and potentially lower-power devices for real-time inference, while requiring only modest additional calibration-time computation. The approach is intended to improve efficiency and hardware feasibility rather than to expand model capability, and it does not require new data collection beyond standard calibration samples. As with many efficiency improvements, easier deployment may warrant additional safeguards to support responsible use; we recommend deploying the method with appropriate access controls and governance practices.

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

# A. Experimental Details

**Baseline Implementation.**    We implement convolutional logic layers via an `im2col`/`unfold`-style transformation. For each image, we extract all $K \times K$ receptive-field patches at every spatial location and flatten them, unfolding patches from all input channels once to form a per-location feature vector.

We follow the original convolutional logic network connectivity: each output channel samples two input channels uniformly at random and keeps this selection fixed. Accordingly, at each spatial location the output-channel computation uses only the two corresponding patch blocks from the unfolded vector and ignores the remaining channels.

This formulation allows us to leverage the public CUDA kernel from the original logic network implementation (for FC-like logic layers) by treating spatial locations as batch elements and sharing the same parameters across all locations (Petersen et al., 2022).

**Model variants.**    We follow the reference convolutional logic network (LogicTreeNet) configurations from (Petersen et al., 2024), where `(S)` and `(M)` denote the Small/Medium width variants. Accordingly, the first convolutional layer has 256/32 output channels for `CIFAR-10(M/S)` and 64/16 output channels for `MNIST(M/S)`; all other architectural settings follow the released implementation.

**Post-Tying Fine-tuning Protocol.**    After completing layer-wise tying across all layers, we perform a single fine-tuning run to recover accuracy. We use the standard training recipe from the convolutional differentiable logic gate network implementation (Petersen et al., 2024): AdamW with weight decay 0.002, learning rate 0.02 (CIFAR-10) or 0.01 (MNIST), for 30,000 iterations with batch size 128. Throughout fine-tuning, we keep the selected ties *fixed* and prevent fine-tuning from changing the tying configuration, so that the tied ratio (and corresponding post-synthesis footprint) remains unchanged. We apply the same fixed-tying protocol for all "w/ FT" results, including baselines, for a fair comparison.

**FPGA Synthesis Setup.**    We synthesize the generated Verilog RTL using Xilinx Vivado (v2021.2) targeting the S2C SingleEv7 platform (V7-2000T). Following (Petersen et al., 2024), we reconstruct a baseline pipeline schedule to match the reported throughput targets and keep it fixed while applying unit tying. Concretely, we insert pipeline registers after each convolution and or-pool layer (with an additional register after the final classifier layer for `MNIST(M/S)`). For the GroupSum reduction, we use a pipelined adder tree with model-specific fan-in partitioning: `CIFAR-10(M/S)` forms 4 partial sums per group and performs a final reduction, whereas `MNIST` forms 64 partial sums and applies an intermediate reduction (16-way for `MNIST(M)` and 8-way for `MNIST(S)`) before the final reduction. We run out-of-context synthesis with the `AreaOptimized_high` directive, and report post-synthesis LUT utilization from Vivado reports.

# B. Detailed Results

**Classification Performance on Small (S) Models.**    Table 7 reports accuracy (mean $\pm$ std) for the Small (S) variants on CIFAR-10 and MNIST across tied ratios. The table complements the main results by showing consistent trends on smaller models, including both post-tying performance and the recovery achieved by a single post-tying fine-tuning run.

*Table 7.* Results for Small (S) models on CIFAR-10 and MNIST.

| Model | Method | 10% Tied | 20% Tied | 30% Tied | 40% Tied | 50% Tied |
|---|---|---|---|---|---|---|
| CIFAR-10(S) | ConstProb | $56.56 \pm 0.00$ | $51.47 \pm 0.00$ | $42.11 \pm 0.00$ | $31.65 \pm 0.00$ | $22.43 \pm 0.00$ |
| (*original*: 58.39) | Fisher-TL | $56.72 \pm 0.04$ | $53.62 \pm 0.70$ | $50.41 \pm 0.69$ | $44.78 \pm 0.69$ | $39.15 \pm 2.16$ |
| | FD-DL | $57.07 \pm 0.17$ | $\mathbf{55.39 \pm 0.08}$ | $52.81 \pm 0.63$ | $46.74 \pm 0.49$ | $38.17 \pm 1.57$ |
| | *Ours (Stage-A only)* | $57.08 \pm 0.21$ | $54.85 \pm 0.17$ | $51.64 \pm 0.15$ | $46.89 \pm 0.80$ | $38.89 \pm 1.19$ |
| | *Ours* | $\mathbf{57.14 \pm 0.30}$ | $55.25 \pm 0.46$ | $\mathbf{52.94 \pm 0.46}$ | $\mathbf{49.50 \pm 0.29}$ | $\mathbf{46.50 \pm 1.10}$ |
| | ConstProb w/ FT | $58.04 \pm 0.06$ | $57.19 \pm 0.16$ | $55.73 \pm 0.13$ | $53.60 \pm 0.18$ | $49.62 \pm 0.17$ |
| | Fisher-TL w/ FT | $57.89 \pm 0.05$ | $57.36 \pm 0.21$ | $56.66 \pm 0.15$ | $55.65 \pm 0.24$ | $54.30 \pm 0.42$ |
| | *Ours* w/ FT | $\mathbf{58.05 \pm 0.15}$ | $\mathbf{57.52 \pm 0.25}$ | $\mathbf{56.70 \pm 0.08}$ | $\mathbf{55.91 \pm 0.28}$ | $\mathbf{54.75 \pm 0.14}$ |
| MNIST(S) | ConstProb | $97.71 \pm 0.00$ | $92.78 \pm 0.00$ | $75.00 \pm 0.00$ | $43.40 \pm 0.00$ | $11.37 \pm 0.00$ |
| (*original*: 98.21) | Fisher-TL | $97.43 \pm 0.11$ | $96.11 \pm 0.44$ | $87.95 \pm 5.99$ | $73.22 \pm 4.05$ | $59.87 \pm 4.22$ |
| | FD-DL | $97.77 \pm 0.05$ | $97.04 \pm 0.14$ | $94.91 \pm 0.31$ | $89.46 \pm 0.91$ | $79.15 \pm 1.17$ |
| | *Ours (Stage-A only)* | $97.70 \pm 0.08$ | $96.77 \pm 0.24$ | $92.73 \pm 0.99$ | $85.09 \pm 1.76$ | $75.10 \pm 3.55$ |
| | *Ours* | $\mathbf{97.84 \pm 0.02}$ | $\mathbf{97.05 \pm 0.07}$ | $\mathbf{95.70 \pm 0.96}$ | $\mathbf{91.15 \pm 0.53}$ | $\mathbf{86.37 \pm 0.84}$ |
| | ConstProb w/ FT | $98.15 \pm 0.03$ | $97.90 \pm 0.09$ | $97.46 \pm 0.14$ | $95.55 \pm 0.05$ | $78.24 \pm 0.40$ |
| | Fisher-TL w/ FT | $98.08 \pm 0.03$ | $97.89 \pm 0.04$ | $97.56 \pm 0.16$ | $97.21 \pm 0.01$ | $96.71 \pm 0.16$ |
| | *Ours* w/ FT | $\mathbf{98.16 \pm 0.06}$ | $\mathbf{98.05 \pm 0.04}$ | $\mathbf{97.84 \pm 0.01}$ | $\mathbf{97.75 \pm 0.09}$ | $\mathbf{97.52 \pm 0.05}$ |

**FPGA Results on Small (S) Models.** Table 8 reports accuracy and post-synthesis LUT usage for the Small (S) models under pixel streaming and the spatially unrolled implementation. Pixel streaming is a resource-efficient implementation that reuses a shared patch-processing datapath and streams patches sequentially. Consistent with the main FPGA results (Sec. 5.4), the Small (S) models exhibit a monotonic decrease in LUT usage as tying increases in both implementations. Overall, unit tying provides a reliable accuracy–area trade-off and helps meet tighter LUT budgets under both implementations.

*Table 8.* FPGA results for Small (S) models across tied ratios. The table compares algorithmic accuracy with post-synthesis FPGA resource usage for Pixel Streaming and Spatially Unrolled implementations. **Red. (%)** denotes the reduction in LUT usage relative to the baseline.

| Model | Tied Ratio | Acc. (%) | Pixel Streaming | | | Spatially Unrolled | | |
|---|---|---|---|---|---|---|---|---|
| | | | Time/Image | LUT (1000s) | Red. (%) | Time/Image | LUT (1000s) | Red. (%) |
| CIFAR-10(S) | 0% tied | 58.39 | $9.22\,\mu s$ | 24.7 | – | 9 ns | 59.3 | – |
| | 10% tied | 57.57 | $9.22\,\mu s$ | 20.6 | 16.73 | 9 ns | 54.3 | 8.43 |
| | 20% tied | 56.88 | $9.22\,\mu s$ | 17.9 | 27.42 | 9 ns | 49.1 | 17.15 |
| | 30% tied | 55.28 | $9.22\,\mu s$ | 15.5 | 37.05 | 9 ns | 41.9 | 29.33 |
| | 40% tied | 53.56 | $9.22\,\mu s$ | 13.6 | 45.13 | 9 ns | 35.3 | 40.49 |
| | 50% tied | 52.22 | $9.22\,\mu s$ | 11.6 | 52.97 | 9 ns | 29.5 | 50.20 |
| MNIST(S) | 0% tied | 98.21 | $3.14\,\mu s$ | 20.0 | – | 4 ns | 26.5 | – |
| | 10% tied | 98.06 | $3.14\,\mu s$ | 16.2 | 18.92 | 4 ns | 22.6 | 14.56 |
| | 20% tied | 97.78 | $3.14\,\mu s$ | 14.3 | 28.45 | 4 ns | 20.2 | 23.57 |
| | 30% tied | 97.85 | $3.14\,\mu s$ | 12.9 | 35.47 | 4 ns | 19.0 | 28.15 |
| | 40% tied | 97.38 | $3.14\,\mu s$ | 11.7 | 41.60 | 4 ns | 16.4 | 38.00 |
| | 50% tied | 96.73 | $3.14\,\mu s$ | 9.6 | 51.75 | 4 ns | 13.3 | 49.95 |

