# OpenReview forum: "Two-Stage Unit Tying for Simplifying Differentiable Logic Gate Networks"
_ICML.cc/2026/Conference — ICML 2026 regular_

### Official Review · Reviewer_C9LT · 2026-03-12

**Soundness:** 3
**Presentation:** 3
**Significance:** 3
**Originality:** 2
**Overall Recommendation:** 3
**Confidence:** 4

**Summary:**

This paper studies post-training simplification for differentiable logic gate networks, with the goal of improving deployability under FPGA resource constraints. The core operation is unit tying, which fixes selected gates to constant 0/1 so that constant propagation and downstream logic elimination can reduce circuit area after synthesis. The authors argue that conventional pruning criteria transfer poorly to this setting because tying is a discrete structural edit rather than a small continuous perturbation. To address this, they introduce a two-stage procedure: a Gauss–Newton-based screening stage under a teacher-referenced logit distortion objective, followed by a finite-difference refinement stage using binary split search to filter out harmful candidates. Experiments on CIFAR-10 and MNIST show improved accuracy–area trade-offs over several baselines, with meaningful post-synthesis LUT reductions.

**Compliance With Llm Reviewing Policy:**

Affirmed.

**Key Questions For Authors:**

1. Sensitivity to model quality before tying. How dependent is the method on starting from a well-trained and well-calibrated teacher model? If the original differentiable logic network is under-trained, does the distortion objective become less reliable?
Choice of overshoot parameter k. The paper includes a sensitivity study, but it would still help to give more practical guidance. Is there a scaling rule relating k to layer width, tying ratio, or calibration set size?

2. Refinement-stage assumptions. The binary split refinement is efficient if harmful candidates are relatively sparse and isolatable. How often does this assumption fail in practice, especially at high tying ratios where interaction effects may be stronger?

3. Effect of post-tying fine-tuning. The results indicate that fine-tuning recovers much of the lost accuracy. How much of the final gain should be attributed to better candidate selection versus simply relying on downstream fine-tuning to repair damage?

4. Structured constraints in convolutional logic layers. The method restricts candidates to the root unit of each logic tree in the convolutional case. This makes sense operationally, but it may also leave potential savings on the table. Can the authors comment on whether finer-grained tying was tested and, if so, why it was not adopted?

**Limitations:**

yes

**Strengths And Weaknesses:**

Strengths
1. Problem formulation is well motivated from the hardware deployment perspective. The paper focuses on a practically relevant bottleneck: differentiable logic networks may be attractive for ultra-low-latency inference, but fitting them into an FPGA budget remains difficult. This makes post-training structural simplification a meaningful systems problem rather than a purely algorithmic exercise.

2. The proposed simplification primitive is hardware-aligned. I find the use of unit tying more compelling than simply borrowing standard magnitude pruning terminology. Tying a gate to a constant enables logic propagation and actual downstream elimination during synthesis, which is more directly connected to post-synthesis area reduction than many abstract model-side sparsification metrics.

3. The method design is coherent. The two-stage decomposition is sensible: a cheap but approximate screening stage followed by a more faithful but selective refinement stage. This is a reasonable compromise between accuracy and search cost, and the paper explains the rationale clearly.

4. The paper evaluates both algorithmic and hardware-level outcomes. It is a positive point that the authors do not stop at pre-synthesis proxies. Reporting synthesized LUT counts and FPGA feasibility makes the claims more convincing, especially for the target application domain.

5. Ablations are useful. The comparisons on calibration size, distortion objective, and refinement strategy help justify the main design choices instead of presenting the method as a monolithic recipe.

Weaknesses
1. Empirical scope is still somewhat narrow. The experiments are limited to CIFAR-10 and MNIST and to a relatively specific family of differentiable logic gate networks. This is enough to establish proof of concept, but not yet enough to show that the method generalizes broadly across logic-network architectures, more challenging datasets, or different task regimes.

2. Comparison set could be stronger. The baselines cover simple constant-probability scoring, Fisher-style task-loss saliency, and finite-difference distortion scoring. These are reasonable, but the paper would be stronger with a broader set of alternatives, especially methods that combine structural simplification with re-training or iterative search procedures, or methods that optimize hardware-aware objectives more directly.

3. The layer-wise greedy protocol introduces confounding factors. Since all methods use the same greedy layer-wise tying pipeline, part of the observed behavior may stem from that procedure rather than from the saliency criterion itself. It would help to discuss more explicitly whether the gain comes mainly from the distortion objective, from the overshoot-and-refine mechanism, or from robustness to greedy interactions across layers.

4. The theoretical analysis is limited. The motivation for why task loss is less discriminative and why distortion-based scoring is preferable is empirically plausible, but the paper remains mostly heuristic. A stronger theoretical account of when the Gauss–Newton approximation should remain informative under near-discrete tying would improve the contribution.

5. Runtime/cost accounting could be more complete.The paper reports some timing in the ablation, but I still do not have a full sense of end-to-end overhead relative to simpler baselines across all experimental settings. Since the method adds a second-stage refinement loop, practical adoption depends on whether the search overhead remains acceptable when model size grows.

6. The hardware evaluation is compelling but still somewhat limited in breadth. LUT reduction is the main reported hardware metric. For FPGA deployment, it would also be useful to see whether tying affects routing pressure, achievable frequency margin, power, or compile/runtime stability beyond the selected examples.

---

> ### Author Rebuttal · Authors · 2026-03-31
>
> We thank the reviewer for the thoughtful and constructive questions. We address each point below with additional experiments and analyses.
>
> ---
> ### **A1. Sensitivity to teacher quality and choice of overshoot k**
>
> We address this question in two parts. First, to test dependence on model quality before tying, we repeated tying using an undertrained teacher checkpoint (65.79% accuracy, compared to 71.57% for the converged teacher). Even in this undertrained-teacher setting, our method still outperforms the task-loss-based Fisher-TL baseline at both tying ratios:
>
> |Method|30% Tied|50% Tied|
> |-|:-:|:-:|
> |Ours|**63.31**|**58.82**|
> |Ours (Stage-A only)|61.73|54.42|
> |FD-DL|62.92|56.07|
> |Fisher-TL|58.34|47.72|
>
> Second, for more practical guidance on the overshoot parameter k, we performed an additional sweep on a larger CIFAR-10(B) model with approximately 2× wider layers than CIFAR-10(M). At the same 50% tying ratio, larger k consistently improved accuracy, indicating that wider models can benefit from a proportionally larger overshoot budget. Because refinement cost also grows with k, our practical takeaway is that k should increase with model width, subject to the desired accuracy-cost trade-off:
>
> |k|Acc.|
> |-|-:|
> |40|68.54|
> |60|69.22|
> |80|**70.32**|
>
> ---
> ### **A2. Refinement-stage assumptions**
>
> Following the reviewer’s insightful suggestion, we examined when the refinement-stage assumption becomes weak by evaluating the accuracy gain from adding refinement across a range of tying ratios, including high tying ratios:
>
> |Stage-A metric|30% Tied|50% Tied|70% Tied|
> |-|:-:|:-:|:-:|
> |Ours|+0.11|+2.22|+7.03|
> |ConstProb|+5.70|+0.29|+0.00|
> |Random|+5.24|+0.11|-0.18|
>
> These results suggest that the refinement assumption is context-dependent; specifically, it depends on how well Stage-A concentrates the overshoot set. Under our logit-MSE based screening, the gain from refinement becomes larger as the tying ratio increases, reaching +7.03 points at 70% tying. This indicates that, for our candidate sets, the remaining harmful units are still sparse enough to be isolated effectively even when interaction effects are stronger.
>
> In contrast, for weak screening baselines such as ConstProb or Random, the refinement gain diminishes rapidly at higher tying ratios. The larger gains at 30% mainly indicate that weaker screening leaves more obviously harmful candidates for Stage-B to remove; this effect quickly vanishes at higher tying ratios. At that point, the candidate sets become noisier and more entangled, leaving much less room for binary-split refinement to recover accuracy.
>
> Overall, these results suggest that the refinement-stage assumption does not fail often for our method, even at high tying ratios, but it can indeed become weak when the screening stage does not produce a sufficiently focused candidate set.
>
> ---
> ### **A3. Effect of Fine-tuning**
>
> We agree that post-tying fine-tuning recovers a substantial portion of the lost accuracy. At the same time, better candidate selection yields a less harmful tying subset and therefore preserves more accuracy in the tied model before fine-tuning. Since all methods are evaluated under the same fixed-tying fine-tuning protocol, fine-tuning cannot change the selected tying subset. Under this controlled comparison, before fine-tuning our method achieves higher accuracy than the baselines, and this advantage is preserved after applying the same fine-tuning budget. Detailed results across different tying ratios are shown in Table 1 of the main paper.
>
> This result suggests that the final outcome depends not only on downstream fine-tuning but also on the quality of the tying subset selected before fine-tuning. From this perspective, better candidate selection and post-tying fine-tuning play complementary roles: better selection reduces the initial damage from tying, while fine-tuning provides additional recovery under the same budget.
>
> ---
> ### **A4. Finer-grained tying**
>
> To explore the potential of finer-grained tying, we evaluated a variant in which all internal units within each convolutional layer were treated as tying candidates under the same layer-wise two-stage procedure.
>
> At 50% tying, this variant achieved 68.63% accuracy with 31.27% LUT reduction. For comparison, the structured root-only design achieved nearly identical accuracy, 68.64%, at a lower tying ratio of 40%, while yielding a larger LUT reduction, 37.82%.
>
> |Variant|Acc.|LUT Red. (%)|
> |-|:-:|:-:|
> |Finer-grained tying|68.63|31.27|
> |Structured root-only tying|68.64|37.82|
>
> We did not adopt the finer-grained design in the main paper because internal-node tying expands the search space substantially (about 7× in convolutional layers) and yields a less structured hardware effect, since it modifies subtrees rather than full output channels. These results suggest that the structured root-only design is more suitable for efficient hardware reduction.

---

### Official Review · Reviewer_vkn8 · 2026-03-12

**Soundness:** 3
**Presentation:** 3
**Significance:** 3
**Originality:** 3
**Overall Recommendation:** 4
**Confidence:** 3

**Summary:**

This paper studies post-training simplification for differentiable logic gate networks, with the
goal of making synthesized circuits fit stricter FPGA capacity budgets. The core idea is “unit
tying”: forcing selected logic units to constants so that constant propagation and synthesis can
eliminate downstream logic. The paper argues that standard pruning criteria transfer poorly to this
setting because tying is a near-discrete structural edit and task-loss changes are often weakly
discriminative in these networks. To address this, the authors propose a two-stage method: a
Gauss–Newton screening stage under a teacher-referenced logit-distortion objective to build an
overshoot candidate set, followed by a binary-split finite-difference refinement stage that removes
harmful candidates. Experiments on CIFAR-10 and MNIST show improved
accuracy–compression trade-offs over several baselines, and hardware synthesis results indicate
substantial LUT reductions that can make previously out-of-capacity designs deployable on
smaller FPGAs.

**Compliance With Llm Reviewing Policy:**

Affirmed.

**Final Justification:**

The rebuttal solved most of my concerns.

**Key Questions For Authors:**

1. How sensitive are the main conclusions to the layer-wise tying schedule and the per-layer
allocation of tie budget T? If a different allocation strategy is used, does the relative advantage
over FD-DL and Fisher-TL remain? A convincing answer would strengthen my confidence that
the gains are not tied to one heuristic schedule.
2. Can the authors provide a small-scale oracle comparison, for example exhaustive or near-
exhaustive search on selected layers or reduced models, to measure how closely Stage-A and the
final two-stage procedure approximate the best tying subset? This would clarify whether the
method is merely better than the baselines or actually close to optimal in representative cases.
3. The paper motivates logit-MSE distortion empirically. Is there a stronger explanation for why
this objective is better aligned with downstream accuracy preservation in tied logic networks than
task loss or KL at moderate temperature? Any additional analysis of rank correlation between
screening scores, FD distortion, and final accuracy would improve the paper.
4. How well does the approach transfer to more challenging datasets or larger logic-network
models, where calibration sets, FD refinement cost, and synthesis effects may behave differently?
Even a discussion of expected bottlenecks or preliminary evidence would help calibrate the scope
of the contribution.

**Limitations:**

See Weaknesses.

**Strengths And Weaknesses:**

--- Strengths

1. The paper is technically coherent and the motivation is well aligned with the
target architecture. The distinction between conventional pruning and tying is
meaningful, and the empirical observations in the motivation section help justify why
off-the-shelf task-loss saliency may be unreliable here. The proposed two-stage design
is also sensible: a cheap second-order screen followed by a higher-fidelity refinement
step is a reasonable way to balance cost and reliability. The empirical results support
the main claim that the full method improves over the chosen baselines on the reported
datasets and models, and the hardware synthesis study strengthens the practical
relevance.

2. The paper is generally clear, well structured, and easy to follow. The motivation
section is particularly effective: the perturbation and loss-landscape figures help explain
why standard pruning intuition may break down. The algorithm description is concise
and readable, and the hardware-deployment discussion makes the practical value
concrete. I also appreciated that the appendix appears to include implementation and
synthesis details, which improves reproducibility.

3. The problem is specialized but important. For this line of work, being able to
trade a moderate amount of accuracy for a large post-synthesis LUT reduction is
practically meaningful, especially because FPGA feasibility is a hard capacity
constraint rather than just a runtime slowdown. The hardware results are therefore a
strong part of the paper. I can see this being useful to researchers working on logic-
centric ML inference and hardware-aware model compression.

4. The paper is meaningfully original in how it reframes post-training simplification
for differentiable logic gate networks. The combination of teacher-referenced logit
distortion, Gauss–Newton screening, and binary-split refinement is not conceptually
radical in isolation, but the adaptation to tying-based logic simplification is well
motivated and seems nontrivial. The most original aspect is the articulation that tying is
a structurally different intervention from conventional pruning and therefore needs
different scoring and refinement behavior.

--- Weaknesses

1. The evidence is still somewhat limited. First, the experimental scope is narrow:
only CIFAR-10 and MNIST are used, on one family of logic-network architectures.
This is enough for a proof of concept, but it leaves open how robust the conclusions are
across harder tasks, different logic-network topologies, or larger-scale settings. Second,
while the paper gives an intuitive argument for why the distortion objective is
preferable, there is no deeper analysis of when the Gauss–Newton approximation
should be expected to correlate with finite-difference distortion under discrete tying.
Third, the refinement step is effective empirically, but its failure modes and dependence
on layer order, tie budget allocation, and candidate interactions are not studied in depth.
Finally, the paper compares against several reasonable saliency baselines, but it would
be stronger to include a more direct oracle-style reference on small layers or a broader
set of structure-aware compression strategies to quantify how much room remains.

2. My main presentation concerns are about precision and completeness rather than
readability. Some claims are phrased a bit strongly, especially “first systematic study,”
without a very sharp delimitation of what related post-training logic simplification work
is excluded. In addition, the paper would benefit from a slightly more explicit end-to-
end description of the layer-wise tying schedule, how T is chosen per layer in the main
experiments, and whether any validation data is used to set calibration size or overshoot
k. A short pseudocode block for the full layer-wise pipeline across the whole network
would help.

3. The broader impact on mainstream ML is more limited. This is not a generally
applicable pruning method for arbitrary neural networks, and the evaluation is not
broad enough to suggest a wider compression principle beyond this niche. So I view the
significance as solid within a focused subcommunity rather than broadly high across
ML.

4. The method is still an incremental algorithmic contribution rather than a major
conceptual breakthrough.

---

> ### Author Rebuttal · Authors · 2026-03-31
>
> We appreciate the reviewer’s careful reading and insightful suggestions.
>
> ---
> ### **A1. Tying schedule and budget allocation**
>
> To address this concern directly, we tested two alternative heuristic pipelines:
>
> **Sensitivity-based schedule.**
>
> We evaluated a sensitivity-based alternative schedule derived from the single-layer tying results, ordering layers from less to more sensitive while keeping the same per-layer tie budgets for all methods.
>
> |Method|30% Tied|50% Tied|
> |-|:-:|:-:|
> |Ours|**66.87**|**56.77**|
> |Ours(Stage-A only)|66.44|55.38|
> |Fisher-TL|59.73|45.63|
> |FD-DL|66.70|52.29|
>
> **Alternative budget allocation.**
>
> We reduced the tying ratio of the last convolutional layer by 10 pp and reallocated that budget to the 2nd and 3rd convolutional layers, while keeping the total tie budget matched to the uniform 30% budget.
>
> |Method|Acc.|
> |-|-|
> |Ours|**64.90**|
> |Ours(Stage-A only)|64.58|
> |FD-DL|64.00|
> |Fisher-TL|59.13|
>
> Across both alternatives, the same qualitative conclusion still holds. This suggests that the gains are not tied to a single heuristic schedule, and that our method remains effective under different layer orders and per-layer budget allocations.
>
> ---
> ### **A2. Small scale oracle comparison**
>
> To better assess how closely Stage-A and the full two-stage method approximate optimal, we compare them against the oracle obtained by exhaustive search. In general, choosing $T$ tied units out of $C$ channels and determining the tie direction for each requires exploring $\binom{C}{T} \cdot 2^T$ possible configurations. Even in a reduced setting with only 32 candidates (the first layer of CIFAR-10(S)), this already amounts to 575,360 cases at $T=4$ and 6,444,032 at $T=5$, illustrating how quickly exhaustive search becomes impractical beyond such a toy setting.
>
> We therefore restrict the oracle comparison to this reduced setting with $T=3,4$. Notably, the full two-stage method exactly matches the oracle at $T=3$, and remains very close to it at $T=4$ with only a 0.25-point gap. In contrast, Stage-A only is 1.82 and 1.81 points below the oracle at $T=3$ and $T=4$, respectively.
>
> |Method|T=3|T=4|
> |-|-|-|
> |Oracle|57.10|56.72|
> |Ours|57.10|56.47|
> |Ours (Stage-A only)|55.28|54.91|
>
> ---
> ### **A3. Why logit-MSE?**
>
> A useful way to view logit-MSE here is that unit tying is a near-discrete edit: many candidates do not immediately change the predicted label, but instead gradually reduce the teacher’s class margins. In this regime, CE can be a weak screening signal because softmax normalization compresses pre-error logit changes, especially on already confident samples. KL is denser than CE, but at moderate temperature it still operates in probability space and therefore inherits part of this compression. By contrast, logit-MSE measures perturbation directly in pre-softmax space, making it better aligned with preserving the teacher’s decision geometry. This is similar in spirit to logit-based knowledge distillation, where matching teacher logits preserves richer information about relative class margins and inter-class structure than relying only on hard labels.
>
> As suggested by the reviewer, we also performed a rank-correlation analysis to measure how well each screening objective aligns with downstream accuracy preservation. Logit-space metrics are better aligned with downstream accuracy than CE, and logit-MSE achieves the highest correlation among the objectives compared here. This gives a more explicit quantitative view of the same phenomenon and supports our use of logit-MSE for Stage-A screening.
>
> |Metric|Spearman|
> |-|:-:|
> |logit-MSE|0.2273|
> |logit-KL|0.2198|
> |CE|0.2077|
>
> ---
> ### **A4. Transfer to challenging datasets and larger models**
>
> We additionally evaluated two model variants to probe transfer beyond the original setting.
>
> - **CIFAR-100**, a more challenging dataset (56.09 original accuracy; 325.7k LUTs)
> - **CIFAR-10(B)**, a larger model with approximately 2× the channel width of CIFAR-10(M) (78.91 original accuracy; 1.01M LUTs)
>
> We observe broadly similar tying and synthesis trends: Ours still outperforms Fisher-TL, remains competitive with FD-DL and maintains a similar accuracy–LUT trade-off.
>
> * Accuracy and LUT Reduction under the same protocol in the main paper
>
> |Model|Method|30% Tied|50% Tied (~44.54% LUT Red.)|
> |-|-|:-:|:-:|
> |CIFAR-10(B)|Ours|**76.04**|**70.32**|
> ||FD-DL|75.71|69.47|
> ||Fisher-TL|74.43|63.99|
>
> |Model|Method|30% Tied|50% Tied (~24.80% LUT Red.)|
> |-|-|:-:|:-:|
> |CIFAR-100|Ours|**55.09**|**51.93**|
> ||FD-DL|54.89|51.47|
> ||Fisher-TL|54.79|49.31|
>
> For the larger CIFAR-10(B) model, increasing the overshoot k improved performance. Even with the increased k (=80), runtime is still only about 400s, leaving our method ~40× faster than FD-DL (about 16,000s) at 30% tying. This suggests that the method remains effective and practical even when scaling to larger models.
>
> Overall, these results provide preliminary evidence that the method transfers well to larger models and challenging datasets.

---

> > ### Author Rebuttal · Reviewer_vkn8 · 2026-04-03
> >
> > The rebuttal substantially improves the paper and addresses several of my earlier concerns. In particular, the added schedule/budget experiments, small-scale oracle comparison, justification for logit-MSE, and transfer results on CIFAR-100 / CIFAR-10(B) all strengthen the empirical case. While I still think the work is more of a strong methodological/engineering contribution than a major theoretical advance, I am now convinced that the paper makes a meaningful contribution to post-training simplification for differentiable logic networks and is supported by sufficient evidence. I therefore update my score to Weak Accept.

---

> > > ### Author Response · Authors · 2026-04-06
> > >
> > > Thank you for your careful consideration  of our rebuttal. We sincerely appreciate your thoughtful evaluation and are grateful that the additional experiments and clarifications were helpful in clarifying several points and strengthening the paper.

---

### Official Review · Reviewer_j6R3 · 2026-03-13

**Soundness:** 3
**Presentation:** 3
**Significance:** 4
**Originality:** 3
**Overall Recommendation:** 5
**Confidence:** 3

**Summary:**

This paper studies post-training simplification of differentiable logic gate networks for FPGA deployment. These models map neural computation directly to gate-level circuits, which allows extremely low latency inference but can produce circuits that exceed FPGA resource budgets. The authors introduce unit tying, which forces selected logic units to constant outputs (0 or 1). This allows constant propagation and removal of downstream logic during synthesis, reducing hardware footprint. The paper argues that conventional pruning criteria are unreliable in this setting because tying introduces discrete structural changes rather than small parameter perturbations. To address this, the authors propose a two stage algorithm. Stage A uses a Gauss–Newton approximation under a teacher referenced logit distortion objective to score candidate units and construct an overshoot set. Stage B refines this set using a binary split search with a small number of finite difference evaluations to remove harmful candidates. Experiments on CIFAR-10 and MNIST show improved accuracy versus compression trade offs compared with several baselines and demonstrate large LUT reductions in FPGA synthesis results.

**Compliance With Llm Reviewing Policy:**

Affirmed.

**Key Questions For Authors:**

* How sensitive is the method to the overshoot parameter k and the calibration set size.
* The experiments focus on CIFAR-10 and MNIST. Do you expect the approach to scale to larger datasets or more complex logic networks?
* Since the distortion objective relies on a teacher network, how sensitive is the method to teacher errors or calibration issues?
* Could tying be integrated with fine tuning during the compression process rather than applied only as a post training step.
* What is the runtime cost of the tying procedure compared with training the original model.

**Limitations:**

yes

**Strengths And Weaknesses:**

### Strengths:
* The paper clearly explains why standard pruning approaches do not transfer well to differentiable logic gate networks. The analysis of discrete tying operations and the flat task loss landscape provides a convincing motivation for a different objective.
* The two stage approach is straightforward and easy to follow. The screening plus refinement structure makes sense computationally since it avoids evaluating expensive finite difference scores for every unit.
* The hardware results are a strong part of the paper. The FPGA synthesis experiments show that tying can reduce LUT usage enough to make models deployable on smaller devices that could not fit the original network.
* The paper includes several ablations that test different components of the method such as the distortion objective and refinement strategy. These help clarify where the gains are coming from.


### Weaknesses:
* The experimental scope is fairly narrow. The evaluation is limited to CIFAR-10 and MNIST with one family of differentiable logic gate networks. It is unclear how well the approach would generalize to other datasets or logic network architectures.
* The models used in the experiments are relatively small. This makes it hard to judge whether the method scales well to larger circuits or more complex tasks.
* The baselines are somewhat limited. Most comparisons are against simple tying heuristics or loss based saliency scores. It would help to see comparisons with stronger or more recent compression approaches adapted to logic networks.
* The theoretical motivation for the Gauss–Newton screening is somewhat heuristic. While the distortion objective is well motivated, the connection between the approximation and the final tying decisions is not fully rigorous.

---

> ### Author Rebuttal · Authors · 2026-03-31
>
> We appriciate the reviewer for the positive assessment and constructive questions. Below we address each point and summarize the additional results.
>
> ---
> ### **A1. Sensitivity to the calibration set size N and overshoot k**
>
> We thank the reviewer for highlighting the importance of this sensitivity analysis. Our evaluation shows that the method is reasonably robust to both the calibration set size N and the overshoot parameter k. For Stage-A, the effect of increasing N quickly saturates, with only marginal change beyond N=16. Stage-B shows a similar behavior, reaching near saturation by around N=20, and changes only marginally thereafter. The overshoot sweep likewise shows a broad plateau for k≥20. For a more comprehensive view of these results, we have included the full visualization of these trends in Appendix C (Fig. 6).
>
> ---
> ### **A2. Larger datasets and more complex models**
>
> We additionally evaluate on CIFAR-100 as a more challenging dataset and on a CIFAR-10(B) model with 2× channel width relative to the main CIFAR-10(M) model, as a more complex logic network. The B model has original accuracy 78.91 and post-synthesis LUT usage of about 1.01M, while the CIFAR-100 model has original accuracy 56.09 and LUT usage of 325.7k.
>
> * Accuracy and LUT Reduction under the same protocol in the main paper:
>
> |Model|Method|30% Tied|50% Tied (~44.54% LUT Red.)|
> |-|-|:-:|:-:|
> |CIFAR-10(B)|Ours|**76.04**|**70.32**|
> ||FD-DL|75.71|69.47|
> ||Fisher-TL|74.43|63.99|
>
> |Model|Method|30% Tied|50% Tied (~24.80% LUT Red.)|
> |-|-|:-:|:-:|
> |CIFAR-100|Ours|**55.09**|**51.93**|
> ||FD-DL|54.89|51.47|
> ||Fisher-TL|54.79|49.31|
>
> Taken together, these results suggest that the method remains effective beyond the original CIFAR-10/MNIST settings: it stays competitive on the larger CIFAR-10(B) model and achieves the best accuracy on the harder CIFAR-100 task, while still providing meaningful post-synthesis LUT reduction.
>
> ---
> ### **A3. Teacher errors and calibration issues**
>
> We agree that sensitivity to both teacher errors and calibration quality is important to assess. To examine these effects separately, we performed two stress tests under the same tying protocol as in the main paper.
>
> **Calibration-stress test**. We constructed a deliberately hard calibration set using only samples misclassified by the teacher, i.e., cases where the teacher signal is least reliable. The results are:
>
> |Method|30% Tied|50% Tied|
> |-|:-:|:-:|
> |Ours|**67.32**|**59.20**|
> |Ours (Stage-A only)|66.15|56.88|
> |FD-DL|67.12|55.69|
> |Fisher-TL|64.25|52.44|
>
> **Teacher-error stress test**. We also repeated tying using an undertrained teacher checkpoint (65.79% accuracy, versus 71.57% for the converged teacher). The results are:
>
> |Method|30% Tied|50% Tied|
> |-|:-:|:-:|
> |Ours|**63.31**|**58.82**|
> |Ours (Stage-A only)|61.73|54.42|
> |FD-DL|62.92|56.07|
> |Fisher-TL|58.34|47.72|
>
> Overall, the method remains reasonably robust under both stress settings. It stays competitive in the harder calibration case and retains a clear advantage at higher tying ratios and under an undertrained teacher.
>
> ---
> ### **A4. FT integrated Tying?**
>
> Yes, tying can be integrated with intermediate fine-tuning during compression. To verify this directly, we tested the following two pipelines at a final tying ratio of 20%:
>
> - One-shot: 20% tying followed by a single 30k-iteration fine-tuning stage
> - Iterative: 10% tying → 30k-iteration fine-tuning → additional 10% tying → 30k-iteration fine-tuning
>
> |Pipeline|Acc.|
> |-|-|
> |One-shot|71.12|
> |Iterative|71.18|
>
> Each fine-tuning stage used the same recipe as in the paper. The iterative variant gives a small additional gain, indicating that the proposed method is compatible with an integrated tie-and-fine-tune pipeline. At the same time, the improvement is modest relative to the additional fine-tuning budget, so we chose the simpler one-shot post-training pipeline in the paper for a cleaner and more cost-efficient evaluation.
>
> ---
> ### **A5. Runtime cost**
>
> For CIFAR-10(M), training the original model from scratch takes about 100k iterations (∼6 hours) to converge. In contrast, the two-stage tying procedure takes within 2 minutes, and the subsequent post-tying fine-tuning uses 30k iterations (~1.8 hours), which is substantially cheaper than retraining from scratch.
>
> More importantly, unlike width reduction with retraining, post-training tying provides a practical hardware-aware control knob. Since post-synthesis LUT usage cannot be predicted accurately from the original model size alone, adjusting width by retraining may require several costly train-and-synthesize rounds before finding a model that just fits the target FPGA budget. In contrast, our method adjusts the model size directly from a trained model, enabling up to 180× faster size control and making it practical to find a just-fit model for a target FPGA budget.

---

> > ### Author Rebuttal · Reviewer_j6R3 · 2026-04-05
> >
> > Thank you for the further studies in addressing the comments. I already have the paper set to accept, I am not changing my score.

---

> > > ### Author Response · Authors · 2026-04-06
> > >
> > > Thank you for your thoughtful comments and for carefully considering our rebuttal. We sincerely appreciate your time, constructive feedback, and positive assessment of our work.

---

### Official Review · Reviewer_LWRF · 2026-03-17

**Soundness:** 3
**Presentation:** 3
**Significance:** 3
**Originality:** 3
**Overall Recommendation:** 4
**Confidence:** 4

**Summary:**

Different from previous work, the paper is viewing the pruning/compression problem as function preservation instead of minimizing the task-loss. They argue that for logic networks the loss landscape is too flat (Fig. 3 shows a specific case in 2 dimensions) and thus makes previous saliency metrics unreliable. They show clearly better pruning/tying to quality/accuracy trade-offs than the baseline methods.

**Compliance With Llm Reviewing Policy:**

Affirmed.

**Final Justification:**

The rebuttal has partially addressed my concerns. Concerns remain around applicability beyond CNNs. It is not a bad paper and has no major technical flaws. It is not the strong paper that definitely needs to be in the program either, though.

**Key Questions For Authors:**

see weaknesses

**Limitations:**

yes

**Strengths And Weaknesses:**

strengths:
- well-motivated new approach, with the 2-stage structure matching the problem well
- clear , informative experiments with nice trade-off analyses
- evaluation with device mapping & optimizations the FPGA tools apply on top.
- the sensitivity study in Appendix C provides confidence that the method isn't brittle and the experiments don't depend on good random seeds.

weaknesses:
- The float loss landscape observation has been shown for a specific network and task/dataset. Does it also hold for non-convolutional models?
- The random search baseline, while a decent sanity check, is quite weak. How does it compare with similar time budget and combined with FD?
- The method greedily goes through the layers -> what about cross-layer interaction? Does re-analysis after a first layer of pruning/tying change the outcome?
- The results show that fine-tuning remains absolutely essential. An obvious variant of the proposed method would be to iteratively tie a smaller share of the weights, then fine-tune, tie the next weights, fine-tune, ... as I would expect saliency to change after tying some of the units. Such an ablation would be helpful to better understand how the saliency evolves.

---

> ### Author Rebuttal · Authors · 2026-03-31
>
> We thank the reviewer for careful reading and constructive suggestions. Below we address each point and summarize the additional analyses.
>
> ### **A1. Loss Landscape**
>
> The same qualitative behavior—namely, a relatively flat task-loss response and a more discriminative logit-distortion response—is also observed across additional DNN variants, including a MLP-style non-convolutional model; representative plots are included in the anonymous supplementary material.
>
> https://anonymous.4open.science/r/TwoStageUnitTying-2823/landscape_comparison.png
>
> ---
> ### **A2. Comparison with stronger random baseline**
>
> The reviewer raises a useful point regarding the strength of the random-search baseline. To address this, we conducted an additional experiment in which, instead of using our screening and refinement stages, we repeatedly sampled random tying subsets, evaluated them using FD as the selection criterion, and selected the best subset under a comparable runtime budget. As shown in the table, our method consistently achieves substantially higher accuracy across all tying ratios with comparable runtime. This provides further evidence that our two-stage procedure, especially the screening stage, is effective at identifying more promising tying subsets within a similar runtime budget.
>
> |Tied Ratio|Random + FD (Acc)|Ours (Acc)|
> |:-:|:-:|:-:|
> |10%|54.64|70.76|
> |20%|31.07|69.52|
> |30%|17.20|67.30|
> |40%|15.88|65.05|
> |50%|10.59|61.44|
>
> ---
> ### **A3. Cross-layer interaction and re-analysis**
>
> To examine whether the greedy layer-wise procedure sufficiently captures cross-layer interaction, we performed two complementary analyses.
>
> **Revisiting the first layer**
>
> We tested whether an earlier layer needs to be re-analyzed after later layers have been tied. After running the standard early-to-late procedure, we re-ran two-stage selection only for the first layer while keeping later-layer ties fixed. Although the selected subset changed moderately, the final accuracy changed very little. This suggests that the early-to-late procedure is already a reasonable approximation in our setting.
>
> |Tied ratio|Δ Acc.|Tying set similarity (Jaccard)|
> |:-:|:-:|:-:|
> |10%|+0.11|0.7844|
> |20%|-0.02|0.8575|
> |30%|-0.11|0.8777|
> |40%|-0.31|0.9083|
> |50%|-0.18|0.9237|
>
> **Sequential vs. simultaneous layer tying**
>
> For conv1→conv2, we compared two ways of choosing the tying subsets. In our sequential procedure, conv2 is tied after conv1 has been tied. In a non-sequential variant, both conv1 and conv2 are tied once on the original untied model before any tying, meaning that the tying subset for each layer is determined independently without reflecting the effect of earlier tying. The sequential procedure achieved 4.13 points higher final accuracy than this independent-selection variant.
>
> ---
> ### **A4. Iterative tying with intermediate fine-tuning**
>
> To assess whether tie-and-fine-tune updates meaningfully change saliency during compression, we compared a one-shot pipeline with an iterative variant at 20% tying point.
>
> - **One-shot**: 20% tying followed by a single fine-tuning stage
> - **Iterative**: 10% tying -> fine-tuning -> additional 10% tying -> fine-tuning
>
> We compared the resulting accuracy at the same final tying ratio of 20%, with a 30k-iteration budget for each fine-tuning step:
>
> |Pipeline|Acc.|
> |:-:|:-:|
> |One-shot|71.12|
> |Iterative|71.18|
>
> Under this setting, iterative tying performs slightly better than the one-shot procedure. However, the final accuracy gain is modest relative to the additional fine-tuning cost, so we regard the iterative tie-and-fine-tune variant as a possible higher-budget extension. We therefore adopt the simple, efficient one-shot pipeline as the default method in the paper.

---

> > ### Author Rebuttal · Reviewer_LWRF · 2026-04-04
> >
> > Thank you for the rebuttal.
> >
> > - A1: The added DNN LogicNet example hints at generality, but doesn't show it thoroughly. The landscape is also considerably noisier, and crucially, the rebuttal only validates the motivating observation (flat task-loss), not the method itself. For DNN-style layers, with every unit being tying candidate, we have a much larger search space and different Jacobian structure; no actual tying performance data for a DNN architecture is provided.
> > - A2: The Random+FD comparison resolves my concerns regarding this baseline.
> > - A3: The cross-layer analysis is adequate.
> > - A4: The iterative vs. one-shot comparison uses equal iteration counts rather than equal total compute, making the +0.06% difference hard to interpret. how does 2x 10% tying compare to 1x 20% tying, considering the overall execution time of the method?

---

> > > ### Author Response · Authors · 2026-04-06
> > >
> > > We thank the reviewer for the thoughtful follow-up. To directly address the remaining concerns, we performed two additional evaluations.
> > >
> > > ---
> > > ### **A1. Tying results in DNN model**
> > >
> > > We agree with the reviewer that the loss-landscape evidence alone is not be sufficient to demonstration for generality. Therefore, we conducted the same tying experiments and ablation studies on a DNN architecture with a larger search space (width of 512,000, representing 4.36× more tying candidates compared to CIFAR-10(M)) and have included the results below.
> > >
> > > **Post tying Accuracy results**
> > > We report post-tying accuracy across 30%–50% tied ratios. the proposed method consistently outperforms Fisher-TL, FD-DL, and ConstProb. These results show that the method itself remains effective even in a DNN-style architecture with a much larger search space. These results show that the method itself remains effective even in a DNN-style architecture with a much larger search space.
> > >
> > > |Method|30% Tied|40% Tied|50% Tied|
> > > |-|:-:|:-:|:-:|
> > > |Ours|**60.97**|**59.85**|**56.49**|
> > > |Ours(Stage-A only)|60.85|59.85|55.82|
> > > |Fisher-TL|60.39|58.75|55.11|
> > > |FD-DL|60.72|59.20|54.98|
> > > |ConstProb|60.47|58.86|54.43|
> > >
> > > **Runtime in larger search space**
> > > We additionally compared wall-clock runtime on the DNN architecture at 50% tied point. Because this experiment operates in a substantially larger all-unit-candidate regime, runtime becomes an important consideration. As the results show, FD-DL exhibits a much more rapid increase in runtime as N increases, whereas our method maintains a substantially better accuracy–runtime trade-off. These results indicate that our two-stage strategy remains effective and scalable even on a DNN architecture with a larger search space.
> > >
> > > |Method|Acc.|Runtime (s)|
> > > |-|:-:|:-:|
> > > |FD-DL (N=16)|51.99|2236|
> > > |FD-DL (N=32)|54.52|4482|
> > > |FD-DL (N=64)|54.98|8937|
> > > |Ours|56.49|146.2|
> > > |Ours (Stage-A only)|55.82|10.96|
> > >
> > > ---
> > > ### **A4. One-shot versus Iterative under the same budget**
> > >
> > > We agree that comparing iterative and one-shot tying under the same overall compute budget provides a fairer comparison. In our previous experiment, the iterative variant did in fact receive a larger fine-tuning budget through repeated tying-and-fine-tuning rounds. This may indicate that iterative tying can benefit from additional fine-tuning when more budget is available, but it also made the original comparison less direct.
> > >
> > > For the CIFAR-10 (M) model in our setting, the total execution time consists of the tying procedure itself and the subsequent fine-tuning (FT) process. Applying the two-stage tying procedure once takes only about 2 minutes, whereas 30,000 FT iterations take approximately 1.8 hours. Therefore, the time overhead of the tying step is negligible relative to fine-tuning, and matching the total FT budget provides a close approximation to matching the overall execution time.
> > >
> > > **Results.**
> > > To address this directly, we conducted an additional comparison in which both one-shot and iterative variants were given the same total FT budget of 30,000 iterations. For iterative tying, the total tying ratio was distributed uniformly across stages; for example, 30% iterative tying was implemented as (10% tying + 10,000 FT iterations) × 3.
> > >
> > > |Tied Ratio|One-shot|Iterative|
> > > |-|:-:|:-:|
> > > |20%|71.12|**71.17**|
> > > |30%|**70.77**|70.30|
> > > |40%|**69.91**|69.38|
> > > |50%|**69.22**|68.31|
> > >
> > > Under this controlled budget, the one-shot variant performs consistently better at higher tying ratios, and even at 20% it achieves nearly identical accuracy to 2×10% iterative tying (71.12 vs. 71.17). Overall, these results suggest that, when the fine-tuning budget is limited, applying the full target tying ratio in a single step is generally more effective than splitting the same budget across multiple smaller tying-and-fine-tuning rounds. This more directly supports our choice of one-shot tying as the default setting in the paper.

---

### Decision · Program_Chairs · 2026-04-30

**Decision:**

Accept (regular)

**Comment:**

Most reviewers are positive, I respect the reviewers. But I also agree with that empirical scope is still somewhat narrow. The experiments are limited to CIFAR-10 and MNIST, even they rebut with a larger CIFAR dataset, but it is better they can have a bigger one different from CIFAR and MNIST. So I vote for weak accept.